# Plant Extracts and Reactive Oxygen Species as Two Counteracting Agents with Anti- and Pro-Obesity Properties

**DOI:** 10.3390/ijms20184556

**Published:** 2019-09-14

**Authors:** Hanna Zielinska-Blizniewska, Przemyslaw Sitarek, Anna Merecz-Sadowska, Katarzyna Malinowska, Karolina Zajdel, Marta Jablonska, Tomasz Sliwinski, Radoslaw Zajdel

**Affiliations:** 1Department of Allergology and Respiratory Rehabilitation, Medical University of Lodz, 90-725 Lodz, Poland; hanna.zielinska-blizniewska@umed.lodz.pl (H.Z.-B.); katarzyna.malinowska@umed.lodz.pl (K.M.); 2Department of Biology and Pharmaceutical Botany, Medical University of Lodz, 90-151 Lodz, Poland; przemyslaw.sitarek@umed.lodz.pl; 3Department of Economic Informatics, University of Lodz, 90-214 Lodz, Poland; marta.jablonska@uni.lodz.pl (M.J.); radoslaw.zajdel@uni.lodz.pl (R.Z.); 4Department of Medical Informatics and Statistics, Medical University of Lodz, 90-645 Lodz, Poland; karolina.smigiel@umed.lodz.pl; 5Laboratory of Medical Genetics, Faculty of Environmental Protection, University of Lodz, 90-236 Lodz, Poland; tomasz.sliwinski@biol.uni.lodz.pl

**Keywords:** obesity, plant extracts, polyphenols, reactive oxygen species

## Abstract

Obesity is a complex disease of great public health significance worldwide: It entails several complications including diabetes mellitus type 2, cardiovascular dysfunction and hypertension, and its prevalence is increasing around the world. The pathogenesis of obesity is closely related to reactive oxygen species. The role of reactive oxygen species as regulatory factors in mitochondrial activity in obese subjects, molecules taking part in inflammation processes linked to excessive size and number of adipocytes, and as agents governing the energy balance in hypothalamus neurons has been examined. Phytotherapy is the traditional form of treating health problems using plant-derived medications. Some plant extracts are known to act as anti-obesity agents and have been screened in in vitro models based on the inhibition of lipid accumulation in 3T3-L1 cells and activity of pancreatic lipase methods and in in vivo high-fat diet-induced obesity rat/mouse models and human models. Plant products may be a good natural alternative for weight management and a source of numerous biologically-active chemicals, including antioxidant polyphenols that can counteract the oxidative stress associated with obesity. This review presents polyphenols as natural complementary therapy, and a good nutritional strategy, for treating obesity without serious side effects.

## 1. Introduction

Plant-based therapy and supplementation are common complementary treatments in various health problems. Several plant compounds may play a unique role in managing a number of diseases, and in vitro and in vivo studies have found some to exert positive effects against obesity. This effect has been attributed to the effects of numerous active compounds on the modulation of various signaling pathways. Thanks to their safety and extended efficacy, plant compounds are regarded as alternatives to traditional pharmaceuticals for treating obesity, or as the basis for the development of new pharmaceutical agents [1].

Despite being preventable, obesity is a growing, multifactorial disease afflicting both developed and developing countries. It is a serious problem for public health, and one that is associated with an increased rate of mortality, morbidity and decreased quality of life. It affects all age groups in populations including children, teenagers and adults. A 2016 WHO study [2] found overweight to affect 1.9 billion (39%) adults worldwide and obesity 650 million (13%); in addition, 41 million children aged under 5 and over 340 million aged 5 to 19 had a weight problem. The worldwide prevalence of obesity has nearly tripled since 1975 and if this trend continues, it is estimated that by 2030, 38% of the population aged over 18 years will be overweight and 20% will be obese [3].

Obesity arises as a result of an energy surplus between calories consumed from food and those expended in metabolism and physical activity, with the correct energy balance being determined by genetic, epigenetic, and environmental factors. Overweight and obesity are most widely determined based on body mass index (BMI), is defined as weight (kg) divided by height in meters squared (m). A BMI range of 18.5–24.9 kg/m^2^ and 18.5–23 kg/m^2^ is considered ideal, while overweight is defined as a BMI of 25 to 29.9 and 23 to 27.5 kg/m^2^, and obesity of ≥30 kg/m^2^ and ≥27.5 kg/m^2^ for Caucasian and Asian population, respectively [4,5,6,7]. BMI tends to be higher among physically active individuals [8] and shorter adults [9] because may relate to increases in muscle mass and inverse BMI–height association, respectively. Additionally, BMI does not acquire body fat location information [10].

Obesity and overweight are defined as a condition of abnormal or excessive fat accumulation in adipose tissue resulting in impaired health. Both are associated with an elevated risk of various diseases, including type 2 diabetes, dyslipidemia, and cardiovascular diseases such as hypertension, stroke, and coronary heart disease. Overweight and obese people are also more subject to sleep apnea, cognitive dysfunction, nonalcoholic fatty liver disease, osteoarthritis, trauma and infection, and infertility. In addition, cancers such as endometrial, breast, prostate, colorectal, esophageal, renal cell, pancreatic, ovarian, thyroid, and gallbladder are also more prevalent [11,12,13].

Although the basis of obesity treatment remains a combination of dietary change and physical activity, this multifactorial, chronic disease requires a multifaced approach. Individuals who fail to respond to lifestyle modifications may require a different therapy; one such approach uses herbal medicine as supportive therapy. The present study reviews in vitro and in vivo studies of selected natural plant extracts believed to have anti-obesity potential and identify potential signaling pathways by which these active compounds may act. 

## 2. Reactive Oxygen Species (ROS) and Polyphenols as Two Opposing Pro and Anti-Obesity Factors

ROS are molecules with an uncoupled electron that mediate cellular signal transduction. In living organisms, ROS are generated in the mitochondria, cytosol, peroxisomes, endoplasmic reticulum (ER), plasma membrane and lysosomes by various cellular systems [14], although most production occurs through the electron transport chain in mitochondria and nicotinamide adenine dinucleotide phosphate (NADPH) oxidases. ROS are divided into reactive oxygen radicals including superoxide (O2^•−^), hydroxyl (^•^OH), peroxyl (RO2^•^), alkoxyl (RO^•^), and non-radicals such as hypochlorous acid (HOCl), ozone (O^3^), singlet oxygen (^1^O2), and hydrogen peroxide (H_2_O_2_) that are converted into radicals [15]. While ROS play important roles in physiological cellular processes, their imbalance and excessive production leads to oxidative stress, followed by damage to biomolecules such as lipids, proteins, and DNA. To counter excessive oxidative stress, the human organism employs a range of antioxidants whose function is to scavenge ROS; these can be produced endogenously, such as superoxide dismutase (SOD), catalase (CAT), and glutathione peroxidase (GPX), or can be obtained exogenously, such as polyphenols [16]. ROS are involved in a number of processes in the human body, including the maintenance of the balance between mitochondria biogenesis and mitophagy, as well as the levels of inflammation, appetite, and energy homeostasis associated with obesity pathogenesis [17,18,19].

Bioactive compounds derived from plants may be biologically active natural products that exert positive health effects [20]. Despite the differences in their chemical structure, and their quantity and location in the plant, all have demonstrated considerable potential in therapies for various diseases, particularly regarding obesity. One of the most widespread groups of plant secondary metabolites is that of the polyphenols. Polyphenols range from simple chemical structures to highly-polymerized molecules, but all are based on a one or more aromatic ring which can be bound to one or more hydroxyl groups, or to other substituents such as methoxyl or carboxyl groups. More than 8000 polyphenols have been isolated from plants, half of which are flavonoids [21].

Polyphenols can be divided according to their chemical structures into phenolic acids, including (caffeic acid or chlorogenic acid), flavonoids, polyphenolic amides (avenanthramides, capsaicinoids) and other polyphenols (resveratrol, curcumin, rosmarinic acid, gingerol, ellagic acid, valoneic acid dilactone, secoisolariciresinol, and matairesinol). The flavonoids can be subdivided into isoflavones (genistein, daidzein), neoflavonoids (dalbergin), chalcones (phloretin, xanthohumol), flavones (apigenin, luteolin), flavonols (quercitin, kaempferol, myricetin), flavanones (hesperetin) and flavanonols (taxifolin), flavanols (catechins and gallocatechins), and proanthocyanidins (oligomeric procyanidins), anthocyanidins (cyanidin) [22]. These compounds in the plant play important roles including take part in metabolic processes, defense mechanisms, act as antimicrobial agents, serve as pigments to attract pollinators and to camouflage the plant itself [23,24,25,26].

Dietary polyphenol also plays a range of important roles in the human metabolism, in particular: antioxidant, anti-inflammatory, antibacterial, anti-cancer, and also anti-obesity effect related to its free radicals scavenging efficiency [27,28].

Most interest in polyphenols is associated with their antioxidant properties known to counteract the effect of ROS and reactive nitrogen species on protein, lipid and DNAs, thus reducing the risk of cancer [29], diabetes mellitus [30] and its complications [31], inflammation [32], cardiovascular disease [33], and neurodegenerative disease [34]. They are known to significantly reduce the levels of ^•^OH, O^2 •−^, NO^•^, or OONO^−^ by acting as electron or hydrogen atom donors and to prevent the formation of more reactive species by deactivating its precursor. Hydroxyl radical are probably the most known form of ROS; these are generated via multiple pathways, including peroxide reduction, mediated in vivo by either Fe^2+^ or Cu^+^ metal ions via a Fenton-type reaction. The chelating properties of polyphenols reduce the rate of ^•^OH production by binding to transition metals such as Fe^2+^ [22,35]. Additionally, the data indicated that polyphenols act also as co-antioxidants with essential vitamins, or their precursors, thus regenerating carotenoids [36] and vitamins C and E [37] and exerting synergistic antioxidant effects. In vitro studies suggest that polyphenols have the ability to inhibit lipoxygenases and cyclooxygenases prevents the release of arachidonic metabolites, an alternative source of non-mitochondrial ROS [38]. They are also able to increase the activity of potent antioxidant enzymes such as SOD, CAT, and GPX [39]. Antioxidant activity is correlated positively with the number of hydroxyl groups bound to the aromatic ring [40].

## 3. The Impact of ROS on Mitochondrial Activity, the Inflammation Process, Hypothalamic Neurons, and Their Role in Obesity

### 3.1. Mitochondrial Activity in Obese Subjects

Mitochondria play central roles in ATP production, and in eukaryotic cells, the major source of ROS is the mitochondrial respiratory chain during the course of ATP production. Excess levels of nutrients results in abnormalities in mitochondrial number, dynamics, and morphology, as well as defects in biogenesis, ROS production and apoptosis. In adipocytes, mitochondrial dysfunction is closely related to metabolic disorders including altered adipogenesis, lipolysis, fatty acid esterification, and adiponectin production, as well as harmful effects of excessive ROS [41]. In obese individuals, mitochondria show less efficient energy generation and reduced fatty acid oxidation [42], abnormal lipid and glucose metabolism [43] and increased susceptibility to apoptosis [44]. An increase in glucose levels results in increased ROS production, which may damage mitochondrial enzymes and other proteins. In addition, the disruption of the uptake and storage of nutrients, together with the insulin signaling pathway managing the accumulation of lipids and free fatty acids leads to the development of metabolic disorders [41]. Furthermore, transcriptional activation of fatty acid biosynthesis and metabolic reprogramming to glycolysis caused by mitochondrial abnormalities and decreased mitochondrial DNA result in significantly greater formation of lipid droplets [45]. Damaged mitochondria can be removed via a process called mitophagy [46] and replaced by new ones through another process closely linked to ROS production [47]. The upregulation of mitochondrial functions in obesity is associated with metabolic alterations, low-grade inflammation [48] and dysregulated responses of the hypothalamic neurons controlling energy homeostasis to changes in glucose level [49].

### 3.2. Chronic Low-Grade Inflammation as a Result of Increased Adipocyte Number

Obesity is associated with chronic inflammation process. Adipose tissue not only stores fat but also plays a cell regulatory function using a complex network of endo, para and autocrine signals that influence the immune system response, among others. Adipocytes secrete several molecules, such as hormones, growth factors, enzymes, cytokines, complement factors and matrix proteins [50]. The anti- and pro-inflammatory proteins secreted from adipose tissue are called adipocytokines. Of these, adiponectin, transforming growth factor beta (TGFβ), interleukin (IL)-IL-10, IL-4, IL-13, and IL-1 receptor antagonist (IL-1Ra), and apelin are anti-inflammatory, whereas tumor necrosis factor-α (TNF-α), IL-6, leptin, visfatin, resistin, angiotensin II, and plasminogen activator inhibitor 1 are proinflammatory. The anti-inflammatory proteins mediate physiological functions while the proinflammatory ones stimulate inflammatory pathways [51]. In cases of obesity, the number and size of the adipocytes are increased, together with the secretion of various pro-inflammatory molecules, thus enhancing the pathological state [52].

Adipogenesis is a complex process in a series of stages by with differentiation of precursor cells, mainly stem cells-derived, to mature adipocytes is mediated by numerous transcription factors, cell-cycle proteins, hormones, and small molecules. Potential redox-sensitive regulation of that pathways is mediated by receptor tyrosine kinases, peroxisome proliferator-activated receptor γ (PPARγ), PPARγ coactivator 1α (PGC-1α), AMP-activated protein kinase (AMPK), and CCAAT/enhancer binding protein β (C/EBPβ) [53]. In general, the studies indicate that ROS are involve in signal transduction and regulation of adipocytes differentiation, but the precise role remain unclear. NADPH oxidase is a crucial source of ROS in preadipocytes [54]. Activated insulin-like growth factor (IGF) receptor is a tyrosine kinase responsible for activation of downstream signaling pathways such as phosphatidylinositol 3-kinase (PI 3-kinase) and Ras-mitogen-activated protein kinase (MAPK) pathway [55]. ROS regulate MAPK activation, major regulator in cell growth and differentiation via oxidative modifications of important signaling proteins and inactivation of MAPK phosphatases [56]. PPARγ is a central regulator of adipogenesis [57] and there is an evidence that Cys285, located in the ligand-binding domain is redox-sensitive [58]. Another adipogenesis-related redox-sensitive signaling molecule is C/EBPβ, also taking part in that process as important regulator. It has been shown that ROS induce disulfide bond formation between Cys296 and Cys143 followed by dimerization of that transcription factor, additionally potentiating its activity [59]. Additionally, oxidative stress is related to increase expression of PPARγ, C/EBPβ [60], and PGC-1α molecules. Therefore, redox imbalance in fat tissue may lead to abnormal mechanisms of adipocyte differentiation and its dysfunction in obesity.

Inflammation is considered as a collection of protective mechanisms developing in response to various factors, such as tissue damage, pathogens, harmful chemical and physical stimuli. Proinflammatory agents activate appropriate receptors and trigger intracellular signaling pathways, such as the mitogen-activated protein kinase (MAPK), nuclear factor kappa-B (NF-κB), and Janus kinase (JAK)-signal transducer, and activator of transcription (STAT) pathways. Two types of inflammatory responses are distinguished: acute or chronic. Acute immune response is terminated after the elimination of the causative agent and the restoration of homeostasis, while chronic inflammation persists for a longer period of time and can be caused by factors that cannot be eliminated [61].

The alternation of immunity as a result of the excess proinflammatory agents generated in cases of obesity is known as chronic low-grade inflammation [52]. This inflammation is associated with various complications, including cardiovascular disease, diabetes mellitus, and cancer [62]. Although the precise relationship between obesity and inflammation is unclear, the strong link is known to exist between inflammation and ROS [63].

A high-fat diet plays an important role in inflammation of the intestine, which is observed during obesity. Overweight people are characterized by different gut microbiome profiles to those of normal weight [64], with the share of Gram-positive bacterial species tending to grow, resulting in higher levels of lipopolysaccharide (LPS) circulating in the intestine. These LPS are able to activate Toll-like receptors (TLR), resulting in the activation of the myeloid differentiation (MyD)-88 factor, tumor necrosis factor receptor-associated factor (TRAF)-6, IL1R-associated kinases (IRAK1, IRAK4) and the IκB kinase complex. That proteins activates various signaling cascades, including the transcription factors NF-κB, extracellular signal-regulated kinase (ERK), MAPK and c-Jun *N*-terminal kinase (JNK), resulting in the stimulation of the production of inflammatory cytokines, chemokines, and adhesion molecules [65]. Saturated fatty acids are the other ligands for the TLR4 receptors [66], that do not directly bind but through fetuin A [67] or production of danger-associated molecular patterns (DAMPs) [68] and ceramides [69].

ROS are also able to activate NF-κB signaling pathways. Hydrogen peroxide (H_2_O_2_) affects the degradation of IκBα, an NF-κB inhibitor, through tyrosine phosphorylation [70]. After DNA binding, numerous enzymes are activated, including NADPH oxidase, cyclooxygenase-2, and arachidonate 5- and 12-lipoxygenases; this potentiates ROS overproduction or releasing nitric oxide synthases and producing reactive nitrogen species in cells potentiates ROS-induced damage [71]. The presence of ROS upregulates the expression of TLR2 and 4, their related metabolic regulators IRF3 and IRF5, and their signature proinflammatory cytokines in human peripheral blood mononuclear cells via MAPK/NF-κB pathways [72]. Excess nutrition oversupplies skeletal muscle cells and induces NF-κB signaling pathways; this results in overproduction of ROS by interfering with mitochondrial function [73].

Inflammasomes are innate immune system receptors and sensors that induce inflammation in response to pathogenic microorganisms and sterile stressors. They contain the nucleotide-binding domain leucine-rich repeat (NLR) protein, the apoptosis-associated speck-like protein containing a CARD adaptor protein (ASC) and caspase-1. After the formation of the protein complex, caspase 1 activate following the pro-inflammatory cytokines such as IL-1β and IL-18. Overexpression of NLR family pyrin domain containing 3 (NLRP3) and caspase-1 has been identified in obese mice [74] and human subjects [75]. ROS is essential for activation of inflammasome. A two-step mechanism has been proposed for the induction of NLRP3 inflammasomes in macrophages. The first is triggered by microbial or endogenous molecules, resulting in upregulation of that complex and pro-IL-1β expression via NF-κB induction [76]. The second is based on signals from numerous different pathways including ion fluxes, lysosomal destabilization, post-translational modifications of NLRP3 and ROS production. K^+^ efflux is initiated by several activators such as ATP and particulate molecules. Accumulation of unsaturated fatty acids, characteristic of obesity, contribute to membrane decomposition followed by their disruption and NLRP3 activation via K^+^ efflux [77]. The endoplasmic reticulum (ER) releases high levels of Ca^2+^ from storage, resulting in mitochondrial dysfunction and the production of mitochondrial ROS. Additionally, Na^+^ influx and Cl^−^ efflux are also reported as important stimuli for the activation of the NLRP3 inflammasome. Also various activators, such as particulate matter, activate the NLRP3 inflammasome through lysosomal destabilization by inducing lysosomal rupture and cathepsin B release. This process can run in relation to K^+^ efflux.

Two very common post-translational modifications of NLRP3, *viz.* phosphorylation and ubiquitination, may also predispose the NLRP3 inflammasome for activation. One of the first identified NLRP3 inflammasome activation agents are ROS [78]. A high fat diet and excess levels of fatty acids can also induce inflammasome in an AMP-activated protein kinase (AMPK)–autophagy–ROS-dependent manner [79]. The details of how this activation proceeds are unclear but one of the components of the inflammasome complex, NIMA-related kinase 7 (NEK7), may act as a ROS sensor [80].

The ER plays an important role in the storage of glucose, protein, lipid metabolism, and calcium ions, and regulates several cellular processes, including inflammation and nutrient metabolism, via the unfolded protein response (UPR) signaling pathway. In numerous tissues and cell types, UPR activation may occur as a result of increased levels of free fatty acids, particularly saturated fatty acids. Moreover, UPR may by activated by changes in the lipid composition of the ER; this can lead to impairments in the activity of the sarco-/endoplasmic reticulum calcium ATPase followed by an imbalance in homeostasis. UPR may be also direct activated by three proteins localized in the ER: double-stranded RNA-dependent protein kinase-like ER kinase (PERK), inositol-requiring 1α (IRE1α) and, possibly, activating transcription factor-6α (ATF6α) [81]. UPR activation results in the activation of proinflammatory genes, including those encoding TNF-α, IL-1β, IL-6, IL-12p40, and cyclooxygenase-2 by primarily activating NF-κB, an important transcription factor of M1 macrophages [82,83]. The NF-κB signaling pathway also regulates the activity of inflammasomes and induces transcriptional expression of NLRP3 [84].

Many reactions housed within the ER generate ROS, particularly ER chaperones and oxidoreductases. Protein disulphide isomerase (PDI) is a multifunctional redox chaperone. PDI catalyzes disulphide bond formation via an isomerization process associated with sequential oxidation and reduction reactions: the mobile arm of PDI opens when oxidized and closes when reduced; the catalytic cycle is then completed by re-oxidation of PDI. The final re-oxidation is performed by several proteins, including flavin adenine dinucleotide (FAD) binding oxidases, oxidized glutathione, glutathione peroxidase 7 and 8, quiescin sulfhydryl oxidase and also protein endoplasmic reticulum oxidoreductin-1 (Ero1); these take part in an oxygen reduction reaction to produce H_2_O_2_. One mechanism of stress-related ROS induction by the ER involving PDI induction helps prevent protein misfolding [85,86,87]. In a study of 3T3-L1 adipocytes in mice fed a high-fat diet for 16 weeks, ER stress was found to be induced by ROS generation mediated by free fatty acids and excessive expression of inflammatory cytokines [88].

Macrophages play an important role in the inflammatory process. Various mechanisms allow the infiltration of immune cells into obese adipose tissue, including adipocyte death, chemotactic regulation, hypoxia, and fatty acids flux [89]. The macrophage population is divided into activated macrophages (M1) and alternatively activated macrophages (M2). The M1 group is activated by interferon-γ alone or in combination with microbial stimuli or cytokines; it takes part in T helper type 1 (Th1) responses responsible for organ-specific autoimmune disorders. In contrast, the M2 group takes part in tissue remodeling, thus participating in the angiogenesis associated with tumor progression. M2 is activated by IL-4, IL-10, IL13, and IL-33. While M1 is a pro-inflammatory group that releases TNF-α, IL-1β, IL-12, and IL-23 upon stimulation whereas M2 is an anti-inflammatory group that secretes IL-10 and TGF-β [90,91,92]. Obesity state is characterized by a change of macrophage phenotype from M2 to M1, which propagates inflammatory cascades [93].

It has also been suggested that adipose tissue exhibits a hypoxic phenotype due to impairment of oxygen delivery [94]. It has been found in obese mice that the inflammatory M1 polarity of macrophages is activated by hypoxia-related genes in an hypoxia-inducible factor 1-alpha (HIF-1α)-dependent or independent manner [95]. The presence of ROS promotes the proinflammatory M1 phenotype; M1 macrophage activation is correlated with dysregulation of the TNFα mediated inflammatory response. The binding of TNFα to the receptor initiates signaling cascades, including MAPK [96] and the IκB (IKK) kinase through NF-κB signaling pathway activation [97]. TNF-α increases ROS production by activating NAD(P)H oxidase, which releases O^2−^ in vascular tissue [98]. Elevated H_2_O_2_ levels also initiate IKK activation and NF-κB signaling [99]. As the M1 proinflammatory phenotype is favored by the activation of the MAPK and NF-κB signaling pathways, their activation, combined with the presence of ROS, may shift the macrophage phenotype toward M1 and propagate the inflammatory cascade [97]. Other specific inflammatory agents also utilize ROS as a member of their signaling cascade [100,101,102]

### 3.3. Regulation of Appetite and Energy Homeostasis

ROS also play an important role in the central nervous system as a signaling molecule. This is particularly true in the hypothalamus, where they are involved in the regulation of food intake and metabolism processes. They achieve this by exerting an effect on different types of neurons, such as proopiomelanocortin (POMC) neurons and agouti-related protein/neuropeptide Y neurons (NPY/AgRP) [18,103]. POMC neurons use glucose as their main fuel and their activation reduces food intake and increases energy expenditure, whereas NPY/AgRP neurons use fatty acids as main fuel and their activation induces increased food intake and decreased energy expenditure [104]. This balance allows the above processes to be regulated. Mitochondrial ROS are present in the hypothalamus. During a positive energy state ROS are produced as a result of glucose, lipid, insulin, and leptin metabolism in POMC and NPY/AgRP neurons via Ca^2+^ influx and mitochondrial activity. Elevated levels of intracellular ROS stimulates POMC neuron activity and decreases that of NPY/AgRP neurons, thus suppressing food intake and increasing energy expenditure. In a negative energy state, the NPY/AgRP neurons are more active; this decreases the level of ROS by blocking its release. Therefore, oxygen production must be balanced to maintain homeostasis. Overproduction of ROS is related to over-activation of the sympathetic nervous system and may be associated with metabolic disorders, including obesity and related disorders, such as diabetes mellitus type 2, hypertension and cardiovascular dysfunction [18,103,105].

## 4. Obesity—In Vivo and In Vitro Plant Extract Studies

### 4.1. Obesity Cellular Models—In Vitro Studies

Adipocytes store fuel for the body in the form of neutral triglycerides, with these being converted into fatty acids and released when required. Although adipose tissue is mainly composed of adipocytes, it also contains pre-adipocytes, macrophages, endothelial cells, fibroblasts, and leucocytes. Adipocytes are divided into two general types: White and brown. White adipocytes are most common and possess a high capacity for energy storage. White and brown adipocytes develop from mesenchymal and skeletal muscle-like lineages, respectively [106,107,108]. Adipose tissue plays a very important role in the regulation of energy homeostasis by secreting adipokines, which take part in lipogenesis and lipolysis. In addition to its role as an energy store, white adipose tissue also plays an important role in mammalian physiology and is regarded as part of the endocrine system.

One of the in vitro screening strategies used in the analysis of the anti-obesity properties of natural plant extracts is based on the inhibition of lipid accumulation in 3T3-L1 pre-adipocyte cells by a quantitative Oil Red O dye method. The enlargement of adipose tissue is preceded by an increase in the number and size of adipocytes formed by adipogenesis from its fibroblastic preadipocyte precursor [109]. This mechanism is extremely important in obesity development. The differentiation process into mature cells involves treatment with several agents after growth arrest, including insulin, synthetic glucocorticoids and phosphodiesterase inhibitor [110]. Oil Red O is a dye that stains lipids and has been used in the quantitative analysis of adipocyte differentiation and intracellular triglyceride content [111].

All experiments analyzed below were preceded by a cell viability test performed following incubation with an appropriate concentration of plant extracts. In each case, no cytotoxic effects were demonstrated. It was found that several active compounds from various plant species are able to block cell differentiation, suppress adipogenesis, and decrease lipid droplet accumulation and triglyceride level. Selected herb extracts with obtained chemical profile known to have anti-obesity properties, confirmed by experiment on 3T3-L1 cells, are given in Table 1.

The anti-obesity properties of natural plant extracts can also be determined by porcine pancreatic lipase (PPL) in vitro activity assay. PPL hydrolyzes triacyloglycerol into diglycerides and subsequently into monoglycerides and free fatty acids, which process enable dietary fat to be directly and efficiently absorbed by the intestine [137]. To determine PPL inhibition, the plant extracts are pre-incubated with the enzyme, p-nitrophenyl butyrate is added as substrate and then the amount of p-nitrophenol released is determined spectrophotometrically [138]. The procedure typically uses Orlistat, a derivative of lipstatin, an inhibitor of gastric and pancreatic lipase which is widely used as antiobesity drug [139]. The anti-lipase activity of selected plant extracts is presented in Table 2 with characteristic, chemical profiles.

The data presented in Table 1 and Table 2 demonstrate the potential value of the tested plant extract in treating obesity, as confirmed by in vitro studies.

### 4.2. Obesity Animal Models—In Vivo Studies

Animal models, typically Sprague–Dawley (SD) rats, Wistar rats and C57BL/6J mice, are commonly used in the assessment of the weight management potential of plant extracts. The subjects are first given the chance to acclimatize to the specified environmental conditions (appropriate temperatures, humidity, and light and dark cycle). Following this, the animals are typically divided into groups, with one group being fed a normal chow diet and the others being fed different variants of high fat diets containing all essential nutrients, vitamins, minerals and a surplus of fat in order to induce obesity. Measurements are taken at the beginning, over the course of the study and at the end. These measurements commonly consist of body weight, visceral fat and organ weight, and blood biochemistry, including total cholesterol and triglyceride level. Negative and positive control groups with the normal and high fat diet may be administered distilled water and orlistat, respectively. Table 3 presents a number of screened natural plants characterized by chemical profiles which may play a role in alleviating obesity in mouse or rat models.

### 4.3. Obesity Human Model—In Vivo Studies.

Numerous plants contain bioactive substances that influence metabolism and fat oxidation. The following section presents selected plant extracts employed in in vivo human study that demonstrate potential in weight management.

An extensively-studied plant due to its anti-obesity properties is *Camellia sinensis* (L.) Kunzte from the Theaceae family; the species is widely used for making tea, the second most widely-consumed drink after water [182]. Basu et al. [183] examined the impact of green tea on body weight, glucose and lipid profile in 35 patients with obesity metabolic syndrome. The patients were divided into three groups: A control group consuming four cups water/day, a green tea group drinking four cups/day beverage and a supplementation group consuming two capsules/day. The study continued for eight weeks. A significant decrease in body weight and BMI were observed in beverage and supplementation groups as a consequence of elevated oxidation and lipolysis. Elsewhere, significant decreases of weight and BMI were also observed among patients consuming 150 mL green tea beverage two times a day for eight and 12 weeks compared to controls administered placebo [184]. The study included 99 subjects with a BMI of 24–35 and age between 18–50.

*Hibiscus sabdariffa* (L.), of the Malvaceae family, is widely grown in many countries and is consumed as a beverage [185]. Clinical trials found 12-week supplementation with *H. sabdariffa* to be associated with a reduction in body weight, BMI and body fat in thirty-six selected volunteers with BMI ≧ 27 (ages 18–65). The controls were treated with placebo consisting of 500 mg starch capsules, while the study group received capsules with 450 mg extract and 50 mg starch [186].

*Phaseolus vulgaris* (L.) of the Fabaceae family is a legume originating from the American continent. Wild species can be divided into various sub-populations depending on specific geographical regions [187]. Sixty slightly overweight subjects (age 20–45) were divided into two groups: The study group receiving one capsule/day containing 445 mg of *P. vulgaris* beans extract and a control receiving placebo. After a 30-day study period, the supplemented group was found to have lost significantly more weight than the control group [188]. Similar results were obtained in a study of 101 people with BMI of 25–40 and ages between 20 and 50 years: in total, 50 subjects were included into the placebo group and 51 into the study group. Concentration of extract was 1000 mg and the duration of the experiments was sixty days. Again, the supplementation group was found to display significantly greater weight loss [189].

*Nigella sativa* (L.), of the Ranunculaceae, originates from Asia but is now widely used plant throughout the word. The seeds have been used to treat various diseases and ailments and as flavoring in food. [190]. A three-month clinical study evaluated the efficacy of *N. sativa* supplementation on body weight changes. Men aged 30–45 with central obesity, i.e., a waist circumference > 90 mm, received two capsules of 750 mg extract twice a day, while a negative control group received matching placebo. Both groups consisted of 39 subjects. The results indicate that extract of *N. sativa* can induce significant body weight loss [191].

*Gynostemma pentaphyllum* (Thunb.) Makino from the Cucurbitaceae family has been widely used in traditional Chinese medicine and as tea, a vegetable and a supplement in Asia, where it is commonly known as *Jiao Gu Lan*. In a 12-week human clinical trials on 80 subjects with a BMI ranging between 25–30, the study group receiving a 450 mg daily dose of *G. pentaphyllum* extract demonstrated significantly lower body weight and BMI compared to controls [192].

*Kaempferia parviflora* (Wall. Ex. Baker) from the Zingiberaceae family is native to Thailand but is widely used in folk medicine in many countries. [193]. Seventy-six healthy subjects (BMI 24–30 kg/m^2^, age 20–65 years) were enrolled to the study and assigned into two groups. The study group received 150 mg of extract per day, whereas the control group was administered placebo. After 12 weeks, the study group demonstrated significantly greater reduction of body fat [194].

*Carum carvi* (L.) from the Apiaceae family is well known medical plant widely used in Asia, Africa and Europe for food as spice [195]. Their property to weight management were evaluate on obese or overweight aerobic-trained female with BMI of 25–39.9 kg/m^2^ and ages between 20 and 55 years. Study and control group consist of 3 persons per group and volunteers obtained 30 mL/day of extract or placebo, respectively. Eating habits and an aerobic training duration 180 min/week were not changed. Participant were examined for 90 days and after then, a significant reduction of body weight has been demonstrated.

*Cissus quadrangularis* (L.), from the Vitaceae family, is widely used medicinally in Asia and Africa [196]. The activity of *C. quadrangularis* formula, Cylaris was tested on 123 overweight and obese participants with BMI > 25 aged between 19–50. Subjects were divided into four groups: Obese with placebo, obese with two daily doses (514 mg each) of Cylaris, obese with formulation and diet (2100–2200 calories/day), and an overweight group with no diet. After eight weeks, the *C. quadrangularis* formulation appeared to be useful in weight reduction, regardless of diet [197].

*Ziziphus jujuba* (Mill.) (Jujube), of the Rhamnaceae family, is commonly used in Chinese medicine, and its fruit, named dazao, is widely consumed in Asian countries [198]. The anti-obesity properties of the extract were evaluated on group of 83 participants aged between 20–57 after a three-month period of experiments. The volunteers were divided into three groups, one with normal weight and BMI 18.5–24.9, the second group included overweight subjects with (BMI 25–29.9) and the third group included obese subjects with BMI over 30. The participants were randomLy administered 5, 15, and 30 g of *Z. jujuba* powder. All participants lost weight, but weight loss was greatest in the second and third groups after administration of the highest analyzed dose [199].

These plant extract appear to be good natural alternatives for obesity treatment.

## 5. Role of Polyphenols in Mitochondrial Activities, Inflammation and Sympathetic Nervous System Activity and Obesity Management via ROS Neutralization

Polyphenols are commonly known as the largest phytochemical molecules with antioxidant properties [200]. Mitochondrial dysfunction, hyperplasia and hypertrophy of adipose tissue linked to chronic low-grade inflammation process, over-activation of the sympathetic nervous system are related to ROS overproduction. The presence of high levels of ROS is known to play an important part in the etiology of obesity and overweight, and largely contributes to the related pathological outcomes. Damaged mitochondria must be replaced by new ones. The balance between autophagy and biogenesis is essential and influences several pathological conditions [201]. Some polyphenols have the ability to activate sirtuin 1 in vitro and are potential inducers of mitochondrial biogenesis via deacetylation-mediated PGC-1α activation [202]. Excess nutrient levels lead to mitochondrial abnormalities and ROS overproduction [203]. Increased adipocyte size and number results in the initiation of the inflammation process; however, polyphenols are able regulate the immunity response by influencing the synthesis of pro-inflammatory factors, such as cytokines, inhibiting TLR and regulating several inflammatory-related pathways, including NF-κB, MAPK, PI3K/AkT, IKK/JNK, and JAK/STAT. They also interfere with immune cell regulation, pro-inflammatory cytokine synthesis, and gene expression. Polyphenols are able to neutralize ROS by donating an electron or hydrogen atom, deactivating its precursor, chelation properties, exerting co-antioxidant activity with essential vitamins, inhibiting the oxidase and arachidonic acid pathways and upregulating SOD, CAT, and GPX enzymes [204,205]. Polyphenols may act as nutraceuticals preventing hypothalamic inflammation and allowing the regulation of energy balance [206].

The in vitro and in vivo studies reviewed above demonstrate that antioxidants, such as polyphenols offer potential in weight reduction and that beneficial dietary strategies may suppress oxidative stress and prevention obesity, related mitochondrial dysfunction, inflammation, and over-activation of the sympathetic nervous system.

## 6. Conclusions

The plant kingdom is a rich source of active components with a range of biological activities; some of which are polyphenols which have demonstrated anti-obesity properties. Obesity is a significant widespread health problem. Currently, most of the world’s population live in countries where overweight and obesity kills more people than underweight. Even nowadays, despite a range of surgical and pharmacotherapeutic treatments, no efficient risk-free weight management treatment exists. Lifestyle modification, change of diet and increased physical activity are currently regarded as the best option. In addition, in vitro and in vivo data indicate that natural plant supplementation may provide increased health expectancy and significant weight loss through ROS neutralization; the latter can be achieved through their antioxidant capacity, and effects on mitochondrial biogenesis, reduction of inflammation and regulation of the sympathetic nervous system.

## Figures and Tables

**Table 1 ijms-20-04556-t001:** Selected plant extracts with anti-obesity properties with characteristic chemical profiles, as confirmed in in vitro 3T3-L1 cells.

Nr	Plant Sources	Family Name	Tissue Sampled	Class/Bioactive Compounds	Concentration of Extract	Ref.
1.	*Ilex paraguariensis* (A.St.-Hil.)	Aquifoliaceae	leaf and unripe fruit	Polyphenols	50–500 µg/mL	[112]
2.	*Panax ginseng* (C.A. Mey.)	Araliaceae	root	Ginsenosides including Rb1, Rb2, Rc, Rd, Re, Rf, Rg1, Rg2, and Rg3	10 µg/mL	[113]
3.	*Aster spathulifolius* (Maxim)	Asteraceae	leaf	Chlorogenic acid, isoquercitrin, luteolin-7-O-rutinoside, 3,4-di-O-caffeoylquinic acid, 3,5-di-O-caffeoylquinic acid, upatilin.	150–250 µg/mL	[114]
4.	*Taraxacum officinale* (Weber ex F. H. Wigg.)	Asteraceae	leaf and root	Caffeic and chlorogenic acids	300–600 µg/mL	[115]
5.	*Pluchea indica* (L.)	Asteraceae	whole plant	Polyphenols	250–1000 µg/mL	[116]
6.	*Oroxylum indicum* (L.) Kurz	Bignoniaceae	fruit pods	Flavonoids, alkaloids, steroids, glycoside, and tannins	50–200 µg/mL	[117]
7.	*Oroxylum indicum* (L.) Kurz	Bignoniaceae	bark	Oroxylin A, chrysin and baicalein	50 µg/mL	[118]
8.	*Cornus kousa* (L.)	Cornaceae	leaf	Cyanidin 3-glucoside, delphinidin 3-glucoside and pelargonidin 3-glucoside	5–100 µg/mL	[119]
9.	*Cyprus rotundus* (L.)	Cyperaceae	root	Flavonoids, tannins, alkaloid, triterpenoids, saponin	125 µg/mL	[120]
10.	*Vaccinium corymbosum* (L.)	Ericaceae	fruit peel	Polyphenols	50–300 µg/mL	[121]
11.	*Clitoria ternatea* (L.)	Fabaceae	flower	Preternatin A3, ternatin B2, ternatin D2, quercetin-3-rutinoside, ternatin D1, kaemferol-3-O-(2-rhamnosyl) rutinoside, delphinidin-3-glucoside, kaemferol-3-O-rutinoside, delphinidin-3-O-(6-O-p-coumaryl)glucoside-pyruvic acid, (+)-catechin 7-O-β-glucoside, syringetin-3-O-glucoside, quercetin triglycoside, and delphinidin derivatives	500–1000 µg/mL	[122]
12.	*Perilla frutescens* (L.)	Lamiaceae	leaf	Eugenyl glucoside, luteolin-7-O-glucoside, apigenin-7-O-β-d-glucuronide, kaempferol-3-O-β-d-glucuronide and rosmarinic acid	100–400 µg/mL	[123]
13.	*Cinnamomum cassia* (Presl.)	Lauraceae	twig	Coumarin (1), 2-hydroxyl cinnamaldehyde (2), cinnamyl alcohol (3), cinnamic acid (4), cinnamaldehyde (5), 2-methoxy cinnamaldehyde (6), and eugenol	100–500 µg/mL	[124]
14.	*Ficus deltoidei* (Jack)	Moraceae	leaf	polyphenols, triterpenoids	150–300 µg/mL	[125]
15.	*Moringa oleifera* (Lam.)	Moringaceae	leaf	Isoquercitrin, chrysin-7-glucoside, and quercitrin	25–400 µg/mL	[126]
16.	*Nelumbo nucifera* (Gaertn.)	Nymphaeaceae	petal	Quercetin and kaempferol glycosides	2–100 µg/mL	[127]
17.	*Nelumbo nucifera* (Gaertn.)	Nymphaeaceae	leaf	Megastigmanes, alkaloids, flavonoids	100 µM	[128]
18.	*Passiflora edulis f. Flavicarpa* (Degener)	Passifloraceae	leaf	Flavonoids, phenolics, triterpenes, alkaloids	130.56 µg/mL	[129]
19.	*Sasa coreana* (Nakai)	Poaceae	leaf	Caffeic acid, isoorientin, orientin, p-coumaric acid, vitexin, isovitexin, ferulic acid, hesperidin, naringin, luteolin	50–100 µg/mL	[130]
20.	*Zea mays* (L.)	Poaceae	purple corn silk	phenolic acids, flavonoids, anthocyanins	125–1000 µg/mL	[131]
21.	*Coix lachryma-jobi* (L.)	Poaceae	seed	Gallic acid, chlorogenic acid, caffeic acid, ferulic acid	5–30 µg/mL	[132]
22.	*Hovenia dulcis* (Thunb.)	Rhamnaceae	fruit	Quercetin	10–100 µg/mL	[133]
23.	*Coffea arabica* (L).	Rubiaceae	fruits: green dry/fresh; yellow dry/fresh; red dry/fresh	Caffeoylquinic acid, chlorogenic acid, caffeic acid	200–1000 µg/mL	[134]
24.	*Paullinia cupana* (Kunth)	Sapindaceae	seed	Polyphenols	150 µg/mL	[135]
25.	*Lycium chinense* (Miller)	Solanaceae	leaf	Chlorogenic acid, kaempferol-3-sophoroside-7-glucoside, kaempferol-3-sophoroside, kaempferol-3-glucoside, kaempferol	200 µg/mL	[136]

**Table 2 ijms-20-04556-t002:** Selected plant extract with anti-obesity properties confirmed by porcine pancreatic lipase (PPL) in vitro activity assay with characteristic chemical profiles.

Nr	Plant Sources	Family Name	Tissue Sampled	Class/Bioactive Compounds	Concentration of Extract	Ref.
1.	*Panax notoginseng* (Burk.)	Araliaceae	root	ginsenoside Rh4, 20(S)-ginsenoside Rg3 and 20(R)-ginsenoside Rg3	200 μg/mL	[140]
2.	*Cryptolepis elegans* (Wall.)	Asclepiadaceae	leaf	Polyphenols	100 µg/mL	[141]
3.	*Cosmos caudatus* (Kunth.)	Asteraceae	leaf	Quercetin-3-rhamnoside, 1-caffeyolquinic acid, catechin, kaempherol, kaempherol glucoside, quercetin, quercetin-3-glucoside, quercetin-O-pentoside, quercetin-rhamnosyl galactoside, quinic acid, monogalloyl glucose, and procyanidin B1	60–1000 µg/mL	[142]
4.	*Oroxylum indicum* (L.) Kurz	Bignoniaceae	fruit pods	flavonoids, alkaloids, steroids, glycosides, and tannins	100–1250 µg/mL	[117]
5.	*Oroxylum indicum (L.) Kurz*	Bignoniaceae	bark	Oroxylin A, chrysin and baicalein	250 µg/mL	[118]
6.	*Moricandia arvensis* (L.)	Brassicaceae	aerial parts	Kaempferol and quercetin	1000–2500 µg/mL	[143]
7.	*Stellaria media* (Linn.) Vill	Caryophylaceae	whole plant	Polyphenols	1000–5000 µg/mL	[144]
8.	*Silene vulgaris* (Moench)	Caryophyllaceae	leaf	Polyphenols	100 µg/mL	[145]
9.	*Garcinia mangostana* (Linn.)	Clusiaceae	pericarp	Polyphenols, terpenoids	3.91–125 µg/mL	[146]
10.	*Momordica charantia* (L.)	Cucurbitaceae	fruit	Hydroxybenzoic acids, hydroxycinnamic acid, flavonol, isoflavonoid, flavanone, hydroxycoumarin	100–400 µg/mL	[147]
11.	*Diospyros kaki* (Thunb.)	Ebenaceae	fruit	Polyphenols	100–200 µg/mL	[148]
12.	*Phyllanthus niruri* (L.)	Euphorbiaceae	whole plant	Polyphenols	500 µg/mL	[149]
13.	*Phyllanthus chamaepeuce* (Ridl.)	Euphorbiaceae	leaf	Polyphenols	100 µg/mL	[141]
14.	*Garcinia vilersiana* (Pierre)	Guttiefrae	leaf	Polyphenols	100 µg/mL	[141]
15.	*Crocus cancellatus subsp. Damascenus* (Herb.)	Iridaceae	stigma	Catechin hydrate, ferulic and caffeic acids	5000 µg/mL	[150]
16.	*Muscari comosum* (L.)	Liliaceae	bulb	Polyphenols		[151]
17.	*Magnolia officinalis* (Rehd. et Wils)	Magnoliaceae	bark	Polyphenols (Honokiol)	200 μg/mL	[140]
18.	*Memecylon edule* (Roxb.)	Melastomataceae	leaf	Polyphenols	100 µg/mL	[141]
19.	*Nelumbo nucifera* (Gaertn.)	Nymphaeaceae	leaf	Megastigmanes, alkaloids, flavonoids	100 µM	[128]
20.	*Passiflora nitida* (Kunth)	Passifloraceae	leaf	Orientin, vitexin, swertisina	100 µg/mL	[152]
21.	*Portulaca oleracea* (L.)	Portulacaceae	leaf	Polyphenols	100 µg/mL	[145]
22.	*Rubus grandifolius* (Lowe)	Rosaceae	fruit and leaf	Polyphenols	300–4450 µg/mL	[153]
23.	*Citrus unshiu* (S. Marcov.)	Rutaceae	peel	Polyphenols	100–200 µg/mL	[148]
24.	*Lycium chinense* (Miller)	Solanaceae	leaf	Chlorogenic acid, kaempferol-3-sophoroside-7-glucoside, kaempferol-3-sophoroside, kaempferol-3-glucoside, kaempferol	10–1000 µg/mL	[136]
25.	*Capsicum annuum* (L.)	Solanaceae	flower	Polyphenols	100–1000 µg/mL	[154]
26.	*Zygophyllum album* (L.)	Zygophyllaceae	Leaf and flower	Polyphenols	50–200 µg/mL	[155]

**Table 3 ijms-20-04556-t003:** Selected plant extracts with anti-obesity properties confirmed in in vivo mouse or rat models.

Nr	Plant Sources	Family Name	Tissue Sampled	Class/Bioactive Compounds	Species of Animals	Concentration of Extract	Duration of Diet	Mechanisms of Action	Ref.
1.	*Salicornia europaea* (L.)	Amaranthaceae	desalted leaves, branches and stems	Trans-ferulic acid, caffeic acid, p-coumaric acid and isorhamnetin-3-β-d-glucoside	Sprague–Dawley (SD) rats	250 and 500 mg/kg	12 weeks	down-regulation the adipogenesis-related gene expression of sterol regulatory element-binding protein 1 (SREBP-1), peroxisome proliferator-activated receptor γ (PPARγ), CCAAT/enhancer binding protein-α (C/EBPα) and fatty acid (FA) synthase	[156]
2.	*Allium fistulosum* (L.)	Amaryllidaceae	dried bulbs and roots	Ferulic acid and quercetin	C57BL/6 J mice	100 mg/kg	6 weeks	attenuation HFD-induced changes in serum leptin and insulin-like growth factor 1 levels, liver expression of AMPK, and adipose tissue expression of uncoupling protein 2 (UCP2)	[157]
3.	*Ilex paraguariensis* (A.St.-Hil.)	Aquifoliaceae	leaf and unripe fruit	Polyphenols	Swiss mice	1 mg/kg	8 weeks	down-regulation the expression of genes that regulate adipogenesis, such as Creb-1and C/EBPα, and the extract up-regulated the expression of genes related to the inhibition of adipogenesis, including Dlk1, Gata2, Gata3, Klf2, Lrp5, Pparγ2, Sfrp1, Tcf7l2, Wnt10b, and Wnt3a	[112]
4.	*Matricaria recutita* (L.)	Asteraceae	flower	Gallic acid, protocatechuic acid, chlorogenic acid, cafeic acid, cafeoylquinic acid, salicylic acid, quercetin, quinic acid derivative, hydroxybenzoic acid-*o*-hexoside, 5,7,4′-trihydroxy-6,3′-imethoxyflavone	Wistar rats	100 mg/kg	6 weeks	protective effect against obesity and oxidative stress: inhibiting effect on intestinal glucose absorption and/or by negatively regulating the studied intracellular mediators such as calcium, hydrogen peroxide and free iron	[158]
5.	*Erigeron breviscapus* (Vant.)	Asteraceae	whole plant	Scutellarin, 3,5-dicaffeoylquinic acid, 1,5-dicaffeoylquinic acid and 4,5-dicaffeoylquinic acid	C57Bl/6 mice	2% (*w*/*w*)	8 weeks	regulation the expressions of Cyp7α1, CD36 and PPAR-γ	[159]
6.	*Artemisia iwayomogi* (Kitam.)	Asteraceae	whole plant	Scopolin, acetophenone glycoside, scopoletin	C57BL/6J mice	0.5% (*w*/*w*)	11 weeks	downregulation of adipogenic transcription factors: PPARγ and C/EBPα and their target genes: CD36, aP2, and FAS; decreased gene expression of proinflammatory cytokines: TNFα, MCP1, IL-6, IFNα, and INFβ in epididymal adipose tissue	[160]
7.	*Aster glehni* (Franchet et Schmidt)	Asteraceae	leaf	Astragalin and kaempferol	C57BL/6J mice	5% (*w*/*w*)	10 weeks	inhibition the expression of PPARγ, C/EBPα, SREBP-1, liver X receptor, and leptin genes in the epididymal adipose tissue	[161]
8.	*Lithospermum erythrorhizon* (Siebold & Zucc.)	Boraginaceae	root	Shikonin derivatives	C57BL/6 mice	0.25%–0.5% (*w*/*w*)	8 weeks	downregulation of genes involved in the adipogenesis pathway	[162]
9.	*Stellaria media* (Linn.) Vill	Caryophylaceae	whole plant	Polyphenols	Swiss albino mice	400 and 900 mg/kg	6 weeks	delay the intestinal absorption of dietary fat and carbohydrate by inhibiting digestive enzymes	[144]
10.	*Momordica cymbalaria* (Hook.)	Cucurbitaceae	fruit	Gallic acid and rutin	C57BL/6 mice	25–50 mg/kg	10 weeks	amelioration insulin resistance in HFD diet fed C57 mice	[163]
11.	*Dioscorea oppositifolia* (L.)	Dioscoreaceae	rhizome	Polyphenols	ICR mice	0.5% (*w*/*w*)	8 weeks	suppression of feeding efficiency and fat absorption	[164]
12.	*Vaccinium macrocarpon* (Aiton)	Ericaceae	fruit	Delphinidin 3-sambubioside, cyanidin 3-lathyroside, rutin, quercitrin, kaempferol robinobioside, myricetin rhamnoside as well as their aglycones, oligomeric flavan3-ol type B and A, catechin, epicatechin and their gallates, hydroxybenzoic and hydroxycinnamic acids, caffeoylquinic acid, a dihydroxybenzoic acid hexoside and feruloylquinic acid	Wistar rats	200 mg/kg	30 days	improve the metabolic profile and reduced oxidative damage and steatosis	[165]
13.	*Vaccinium corymbosum* (L.)	Ericaceae	fruit peel	Polyphenols	SD rats	60–150 mg/kg	5 weeks	down-regulation of C/EBPβ, C/EBPα, and PPARγ and the reduction of the phospho-Akt adipogenic factor in 3T3-L1 cells	[121]
14.	*Orthosiphon stamineus* (Benth.)	Lamiaceae	leaf	rosmarinic acid	C57BL/6J mice	200 and 400 mg/kg	8 weeks	impact on lipid metabolism	[166]
15.	*Perilla frutescens* (L.)	Lamiaceae	leaf	Eugenyl glucoside, luteolin-7-O-glucoside, apigenin-7-O-β-d-glucuronide, kaempferol-3-O-β-d-glucuronide and rosmarinic acid	C57BL/6J mice	100 and 400 mg/kg	12 weeks	downregulation adipogenic gene and upregulating lipolytic gene expressions	[123]
16.	*Cassia tora* (L.)	Leguminosae	seed	Emodin, aloe-emodin	Wistar rats	100–300 mg/kg	8 weeks	attenuation lipid accumulation in white adipose tissue via AMPKsignaling pathway activation	[167]
17.	*Punica granatum* (L.)	Lythraceae	leaf	Polyphenols	ICR mice	400–800 mg/kg	5 weeks	suppression energy intake	[168]
18.	*Hiptage madablota* (Gaertn.)	Malpighiaceae	root	Terpenoids, polyphenols	Wistar rats	100–400 mg/	40 days	hypophagic and hypolipidemic effects and provoke the brain serotonin level	[169]
19.	*Morus alba* (L.)	Moraceae	root-bark	Kuwanon G, and Albanin G	C57BL/6J mice	250 and 500 mg/kg	7 weeks	appetite control	[170]
20.	*Morus alba* (L.)	Moraceae	leaf	Neochlorogenic acid, cryptochlorogenic, chlorogenic, rutin, isoquercitrin, astragalin acid, nicotiflorin, and protocatechuic acid	Wistar rats	0.5%–2% (*w*/*w*)	4 weeks	regulation adipocytokines, inflammation and oxidative stress	[171]
21.	*Morus alba* (L.)	Moraceae	leaf and fruit	1-deoxynojirimycin, cyanidin-3-glucoside, rutin and resveratrol	C57BL/6 mice	67–167 mg/kg	12 weeks	modulation of obesity-induced inflammation and oxidative stress	[172]
22.	*Moringa oleifera* (Lam.)	Moringaceae	leaf	Isoquercitrin, chrysin-7-glucoside, and quercitrin	C57BL/6J mice	125–500 mg/kg	14 weeks	downregulation the expression of adipogenesis-associated proteins: (PPARγ, C/EBPα and C/EBPβ), and fatty acid synthase (FAS); increased the degree of phosphorylation of AMP-activated protein kinase α (AMPKα) and acetyl-CoA carboxylase (ACC)	[126]
23.	*Nelumbo nucifera* (Gaertn.)	Nelumbonaceae	seed	Polyphenols	SD rats	400 mg/kg	7 weeks	decrease expression of PPARγ, GLUT4, and leptin in cultured human adipocytes	[173]
24.	*Nelumbo nucifera* (Gaertn.)	Nelumbonaceae	leaf	Polyphenols	Wistar rats	70–280 mg/kg	8 weeks	reduction the lipid components	[174]
25.	*Olea europaea* (L.)	Oleaceae	leaf	Oleuropein	C57BL/6N mice	0.15% (*w*/*w*)	8 weeks	reversion the HFD-induced upregulation of WNT10b- and galanin-mediated signaling molecules and key adipogenic genes (PPARγ, C/EBPα, CD36, FAS, and leptin) moreover downregulation of thermogenic genes involved in uncoupled respiration: SIRT1, peroxisome proliferator-activated receptor gamma, coactivator 1 alpha (PGC1α), and UCP1; and mitochondrial biogenesis: transcription factor A, mitochondrial, nuclear respiratory factor-1, and cyclooxygenase-2) was also reversed	[175]
26.	*Passiflora nitida* (Kunth)	Passifloraceae	leaf	Orientin, vitexin, swertisina	SD rats	100 mg/kg	4 weeks	reduction of lipid absorption and pancreatic lipase inhibition	[162]
27.	*Limonium tetragonum* (Thunb.)	Plumbaginaceae	aerial part	(−)-epigallocatechin-3-(3″-O-methyl) gallate, (−)-epigallocatechin-3-gallate, and myricetin-3-O-β-D-galactopyranoside	C57BL/6J mice	100 mg/kg	8 weeks	suppression of adipogenesis-related transcription factors including PPARγ, C/EBPα, SREBP-1 and adipocyte-specific proteins such as fatty acid synthase (FAS), lipoprotein lipase (LPL), and adipocyte fatty acid-binding protein (aP2)	[176]
28.	*Morinda citrifolia* (L.)	Rubiaceae	leaf	Rutin	SD rats	150 and 350 mg/kg	12 weeks	positive influence on the lipid profiles and a reduction in LDL levels	[177]
29.	*Aegle marmelos* (L.)	Rutaceae	leaf	Umbelliferone, esculetin	SD rats	30 mg/kg	2 weeks	counteract the obesity by lipolysis in adipocytes	[178]
30.	*Nephelium lappaceum* (L.)	Sapindaceae	fruit	Geraniin	SD rats	10 and 50 mg/kg	4 weeks	restore the oxidative stress observed in the HFD rats	[179]
31.	*Camellia oleifera* (C.Abel)	Theaceae	fruit	Gallic acid, ellagic acid, 3-O-methylellagic acid 4′-O-β-D-glucopyranoside	ICR mice	100–300 mg/kg	30 days	inhibition fatty acid synthase activity	[180]
32.	*Alpinia officinarum* (Hance)	Zingiberaceae	whole plant	Galangin	C57BL/6J mice	0.5% (*w*/*w*)	8 weeks	suppression protein expressions of C/EBPα, fatty acid synthase, SREBP-1, and PPARγ in the liver and adipose tissue	[181]

The value of plant extract in treating obesity, as confirmed by in vivo studies, are presented in Table 3.

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
