# Peer review of "Plant Extracts and Reactive Oxygen Species as Two Counteracting Agents with Anti- and Pro-Obesity Properties"

_ijms, 2019, doi:10.3390/ijms20184556_

Round 1

Reviewer 1 Report

The aim of the proposed review manuscript by Zielinska-Blizniewska at co-authors was to summarize the current knowledge on the effects of plant-derived bioactive compounds and generation of reactive oxygen species (ROS) on obesity. The authors discussed in details plants bioactive compounds classification, the impact of ROS on mitochondrial stress, inalammation and obesity, and summarized the in vitro and in vivo effects of selected plant extracts with anti-obesity properties.

In the current manuscript, the authors made a good effort to include broad spectrum from the current literature on the effect of plant-derived bioactive compounds on obesity and the counteracting role of ROS. The manuscript is logical and written clearly.

Specific comments and recommendations:

I would encourage the authors, when discussing BMI classification, to discuss briefly the limitations of this classification (for example, physically fit or very short individuals can hardly fit into the healthy BMI category; additionally, BMI criteria for Asian individuals differ from those for Caucasians). When discussing chronic low-grade inflammation as a result of increased adipocyte number, although later in the manuscript discussed, the authors did not distinguish the role of adipocyte maturation (the difference in the secretory activity of the preadipocytes vs. the mature adipocytes) as well as the adipose tissue macrophages which play a major role in the inflammatory process.

In brief, in my opinion, this is a well-written manuscript that would be of interest to the scientific community working in this field.

Author Response

Reviewer 1:
Comments:
“I would encourage the authors, when discussing BMI classification, to discuss briefly the limitations of this classification (for example, physically fit or very short individuals can hardly fit into the healthy BMI category; additionally, BMI criteria for Asian individuals differ from those for Caucasians). When discussing chronic low-grade inflammation as a result of increased adipocyte number, although later in the manuscript discussed, the authors did not distinguish the role of adipocyte maturation (the difference in the secretory activity of the preadipocytes vs. the mature adipocytes) as well as the adipose tissue macrophages which play a major role in the inflammatory process.”

Response:
According to the suggestions on the lines 54-58 information about limitation of BMI classification and BMI criteria for Asian individuals were added. We have discussed the relation of adipocyte differentiation, ROS and obesity on the lines 166-185. The role of adipose tissue macrophages in the inflammatory process were described on the lines 274-297.

Yours sincerely,

Anna Merecz-Sadowska, corresponding author

Reviewer 2 Report

The manuscript by Zielinska-Blizniewska et al. entitled: “Plant extracts and reactive oxygen species as two counteracting agents with anti- and pro-obesity properties” presents a very comprehensive review paper about potential role of plant medicines for the treatment of obesity. The used references are up to date. The manuscript is clearly written. The paper would be interesting for wide range of readers.  

Please find below my minor comments:

Line 6, 3 should be in superscript. Line 279: ‘POMC’ instead of ‘POMOC’. Lines 287-297 incorrect line spacing.

Author Response

Dear Reviewer,

The authors are grateful for analysis of our manuscript and useful suggestions. We appreciate all the insightful comments of the Reviewer.

According to the suggestions we have now changed incorrections in lines 6, 279 and 287-297.

Yours sincerely,

Anna Merecz-Sadowska, corresponding author

Reviewer 3 Report

Manuscript is well written and topic is presented in detail. However, authors should check their manuscript for errors like font type (section 4.1; references ...). Also, tables should be carefully checked (like concentration is written separated; range of concentration should be written in the same way ...)

Author Response

Dear Reviewer,

The authors are grateful for analysis of our manuscript and useful suggestions. We appreciate all the insightful comments of the Reviewer.

According to the suggestions we have now changed incorrections like font type. We repeatedly checked all tables.

Yours sincerely,

Anna Merecz-Sadowska, corresponding author
